# Evaluating the Sustainable Use of Saline Water Irrigation on Soil Water-Salt Content and Grain Yield under Subsurface Drainage Condition

**Genxiang Feng [1,2,]\***, **Zhanyu Zhang [2]** and **Zemin Zhang [2]**

1   College of Water Conservancy and Hydropower Engineering, Hohai University, Nanjing 210098, China
2   College of Agricultural Engineering, Hohai University, Nanjing 210098, China; zhanyu@hhu.edu.cn (Z.Z.); zemin_zhang@outlook.com (Z.Z.)
\*   Correspondence: fenggxhhu@foxmail.com

**Abstract:** A sustainable irrigation system is known to improve the farmland soil water-salt environment and increase crop yields. However, the sustainable use of saline irrigation water under proper drainage measures still needs further study. In this study, a two-year experiment was performed to assess the sustainable effects of saline water irrigation under subsurface drainage condition. A coupled model consisting of the HYDRUS-2D model and EPIC module was used to investigate the effects of irrigation water salinity (IWS) and subsurface drainage depth (SDD) on soil water-salt content and summer maize yield when saline water was adopted for irrigation under different subsurface drainage measures. Summer maize in the two-year experiments were irrigated with saline water of three different salinity levels (0.78, 3.75, and 6.25 dS m$^{-1}$) under three different drainage conditions (no subsurface drainage, drain depth of 80 cm, and drain depth of 120 cm). The field observed data such as soil water content, soil salinity within root zone, ET and grain yield in 2016 and 2017 were used for calibration and validation, respectively. The calibration and validation results indicated that there was good correlation between the field measured data and the HYDRUS-EPIC model simulated data, where RMSE, NSE ($> 0.50$), and R$^2$ ($> 0.70$) satisfied the requirements of model accuracy. Based on a seven $\times$ seven (IWS $\times$ SDD) scenario simulation, the effects of IWS and SDD on summer maize relative grain yield and water use efficiency (WUE) were evaluated in the form of a contour map; the relative grain yield and WUE obtained peak values when drain depth was around 100 cm, where the relative yield of summer maize was about 0.82 and 0.53 at IWS of 8 and 12 dS m$^{-1}$, and the mean WUE was 1.66 kg m$^{-3}$. The proper IWS under subsurface drainage systems was also optimized by the scenario simulation results; the summer maize relative yield was still about 0.80 even when the IWS was as high as 8.61 dS m$^{-1}$. In summary, subsurface drainage measures may provide important support for the sustainable utilization of saline water in irrigation. Moreover, the coupled HYDRUS-EPIC model should be a beneficial tool to evaluate future sustainability of the irrigation system.

**Keywords:** saline water irrigation; subsurface drainage; soil water-salt content; grain yield; coupled model

## 1. Introduction

The lack of fresh-water resources for agricultural irrigation is limiting the sustainable development of agriculture worldwide. In recent years, as a supplemental irrigation water source, saline water has been widely utilized in fresh water-deficient areas. Previous researches have reported that saline water irrigation is acceptable for moderate and mild salt-tolerant crops [1–3]. The response mechanism of soil

salinity and crop yield to saline water irrigation has been studied extensively. However, the strategy to eliminate soil salts added by saline water still need further study [4,5]. Verma et al. confirmed that the salt accumulated in soil profile may lead to the risk of soil salinization [6]. Similar research also found that the decrease of water available for plant root extraction was mainly due to the increase of soil salinity [7,8]. Therefore, in order to ameliorate the soil environment and keep healthy crop growth, appropriate field management practices, including field drainage, and selection of appropriate irrigation water salinity are essential when saline water is utilized for irrigation.

As a subterranean land drainage technology, subsurface drainage was traditionally used for field groundwater management [9–11]. Sharma and Feng both proposed that subsurface drainage could provide great potential for saline water irrigation [12,13]. Extensive research showed that subsurface drainage depth (SDD) was a main factor that affects drainage capacity [14,15]. However, the appropriate drainage depth varies with the change of soil properties and groundwater conditions, it is still in dispute about the appropriate depth [16,17]. Thus, more research is essential to determine the proper drain depth and reveal the effects of subsurface drainage depth on soil water-salt and grain yield, especially in saline water irrigation system [18].

Relying on field experiments alone to determine the proper water management strategies and drainage practices might be incomplete and time-consuming [19]. The HYDRUS model has been adopted in various soil water-salt studies. The model can simulate two- or three-dimensional water and solute transport processes in subsurface drainage systems with its various boundary conditions and sophisticated, graphical, user-friendly interface [20–28]. Since the crop growth response was not considered in the original version of HYDRUS-2D, this model was usually coupled with other crop growth modules such as EPIC, and SWAP to study both the hydrologic and crop growth processes [29]. Among the crop growth modules, EPIC has been a priority used model in crop growth and yield simulation due to a relatively less data input requirement and abundant crop variety options [30,31]. Simulation results by Wang, et al., Han et al. and Wang et al. suggested that the coupled HYDRUS-EPIC model could be used in studying the effects of different agronomic measures such as sprinkler irrigation, saline water irrigation on soil water, solute transport, ET, and grain yield [32–34]. Previous studies using the model have focused more on the effects of irrigation methods, irrigation amount and water salinity on soil water-salt dynamic, grain yield and water use efficiency in different regions worldwide, and the relationships between irrigation and grain yield has been established under various irrigation systems [35,36]. While under drainage condition, especially in subsurface drainage condition, the knowledge of the saline water irrigation still needs further study [37,38]. Together with the field experiment and coupled model, the sustainable effects of saline water irrigation under subsurface drainage condition would be systematically evaluated in various irrigation and drainage scenarios [39,40].

The field and model experiment conducted in this research were: (1) To evaluate the accuracy of the HYDRUS-EPIC model by field experiment data and simulation data; (2) to evaluate the effects of IWS and SDD on summer maize relative yield and WUE by scenario simulation; (3) to determine proper irrigation water salinity under subsurface drainage conditions.

## 2. Materials and Methods

### 2.1. Field Experiments

A two-year experiment was performed in lysimeters during the year of 2016 and 2017 at the Key Laboratory of Efficient Irrigation-Drainage and Agricultural Soil-Water Environment in Nanjing, China (118°60′E, 31°86′N). The mean annual temperature in the experiment area was 15.7 °C, and the mean annual rainfall and evaporation were 1021.3 and 2000 mm, respectively. The summer maize (Longping-206) in this study were irrigated with saline water in three concentration gradients (S1: 0.78, S2: 3.75, and S3: 6.25 dS m$^{-1}$) under three different drainage conditions (D0: no subsurface drainage, D1: drain depth of 80 cm, and D2: drain depth of 120 cm) in three replicates. Summer maize

was sown in mid-June (20 June 2016 and 18 June 20 2017) with the row spacing and plant spacing of 40 cm × 40 cm. The fertilizer application amount was similar to the study by Feng [13]. A total of 225 kg ha$^{-1}$ was applied as base before sowing in the form of ammonium nitrate where P2O5 and K2O at a rate of (N:P2O5:K2O = 28%:6%:6%). Urea was applied at a rate of 300 kg ha$^{-1}$ with the first irrigation water in all treatments during the maize reviving stage. Moreover, the design of the lysimeters was similar to the ones used in Feng's study as shown in Figure 1, where the lysimeters were approximately 15 m$^3$ (2.5 × 2 × 3 m). There were all together 27 lysimeters adopted in this study.

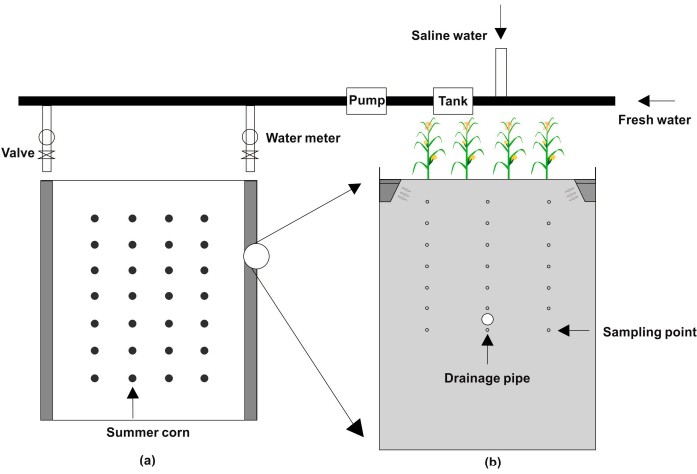

**Figure 1.** (**a**) The layout of the lysimeter; (**b**) the profile map of the lysimeter.

The soil physical properties for different layers (0~20, 20~40, 40~80, and 80~120 cm) in the lysimeters are listed in Table 1. The soil particle percent, soil bulk density, and field capacity were used for estimating initial values of soil hydraulic parameters according to van-Genuchten's model [41]. The estimated soil hydraulic parameters are shown in Table 2.

**Table 1.** Soil physical properties for layered depth.

| Soil Depth (cm) | Soil Particle Percent (%) | | | Soil Bulk Density (g cm$^{-3}$) | Field Capacity (cm$^3$ cm$^{-3}$) |
|---|---|---|---|---|---|
| | Sand (2.0~0.02 mm) | Silt (0.02~0.002 m) | Clay (<0.002 mm) | | |
| 0~20 | 43.91 | 36.41 | 19.68 | 1.33 | 0.38 |
| 20~40 | 42.59 | 37.28 | 20.13 | 1.41 | 0.37 |
| 40~80 | 41.34 | 36.68 | 21.98 | 1.46 | 0.35 |
| 80~120 | 40.25 | 38.12 | 21.63 | 1.48 | 0.23 |

**Table 2.** Initial and calibrated values of soil hydraulic parameters for different soil layers.

| Soil Depth (cm) | $\theta_r$ (cm$^3$ cm$^{-3}$) | $\theta_s$ (cm$^3$ cm$^{-3}$) | $\alpha$ (cm$^{-1}$) | $n$ | $Ks$ (mm d$^{-1}$) |
|---|---|---|---|---|---|
| Initial values | | | | | |
| 0~20 | 0.096 | 0.483 | 0.014 | 1.375 | 149.70 |
| 20~40 | 0.092 | 0.458 | 0.013 | 1.377 | 93.30 |
| 40~80 | 0.089 | 0.441 | 0.013 | 1.372 | 71.20 |
| 80~120 | 0.088 | 0.434 | 0.013 | 1.350 | 62.10 |
| Calibrated values | | | | | |
| 0~20 | 0.096 | 0.483 | 0.015 | 1.200 | 149.71 |
| 20~40 | 0.092 | 0.458 | 0.012 | 1.200 | 93.32 |
| 40~80 | 0.089 | 0.441 | 0.010 | 1.180 | 71.24 |
| 80~120 | 0.088 | 0.434 | 0.012 | 1.252 | 62.12 |

Irrigation water controlled by a water meter was applied to each lysimeter by furrow irrigation. During the summer maize growing seasons in 2016 and 2017, a total of 460 mm irrigation water was applied to each lysimeter in seven irrigation events. Fresh water with irrigation amounts of 60 mm was applied at sowing. In the critical growth periods of the summer maize such as the tasseling and filling stage, two irrigation events with a total irrigation amounts of 120 mm were applied for soil salt leaching. For the other four irrigation events, all irrigation amounts were 40 mm at 10, 62, 92, and 100 days after sowing. The two-year experiments were carried out under a rain shelter. Meteorological data were recorded by an automated weather station located at the experiment field. The maximum air temperature, minimum air temperature, and the irrigation amounts are shown in Figure 2.

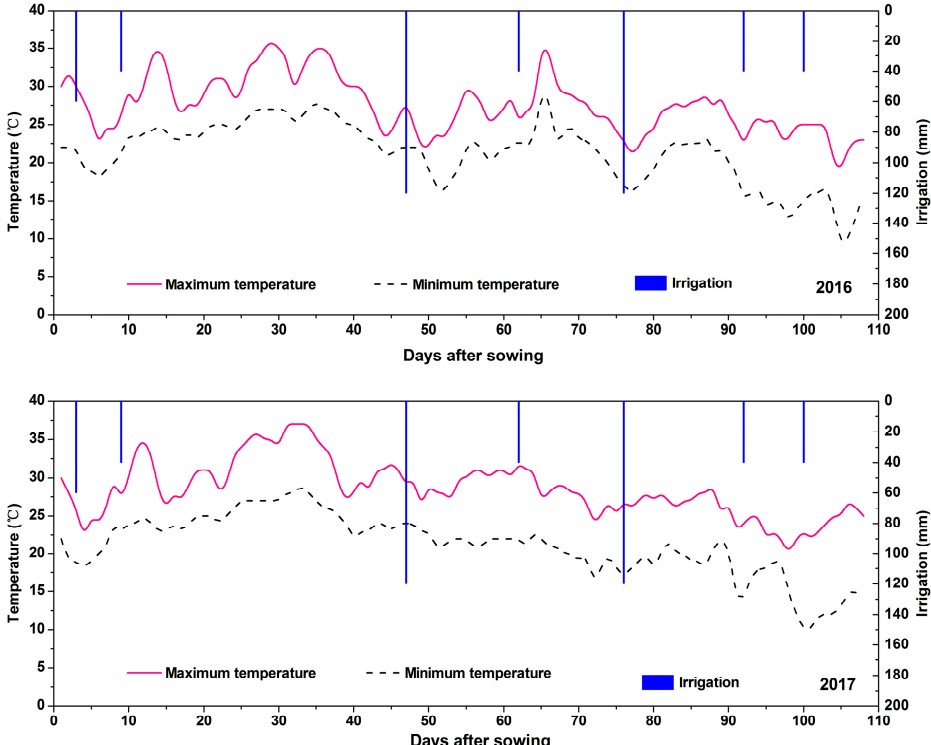

**Figure 2.** Daily values of temperature, and irrigation events during 2016 and 2017 growing seasons.

Soil samples were obtained using a 4 cm auger from soil depths of 20, 40, 60, 80, 100, and 120 cm in each lysimeter, and the sampling interval was 15 days during the summer maize growth period in both 2016 and 2017. The soil water contents were determined using the gravimetric method. $EC_{1:5}$ (dS m$^{-1}$), which is the electrical conductivity of a soil and water mixture in the ratio 1:5 by mass, was determined using an electrical conductivity meter. The $EC_{1:5}$ was then converted to the $EC_e$ (dS m$^{-1}$), which is the electrical conductivity of the saturated paste according to Equation (1) obtained in the laboratory experiment by Feng with $R^2 = 0.994$ [13].

$$EC_e = 14.77EC_{1:5} - 0.225 \tag{1}$$

### 2.2. Model Experiments

The hydrologic module in HYDRUS model was coupled by the crop growth module in EPIC in this study. The crop growth module of EPIC was preferred at field production level due to its less demanding data input requirement, which used a unified approach to simulate more than 100 types of crops [42]. The data exchange between the HYDRUS and EPIC models took "day" as the time step. The coupling model used the hydrologic model in HYDRUS to simulate the soil water and salt dynamics and the effect of soil water-salt on the root water absorption. Then, the results were

output to the EPIC crop growth module to analyze the crop growth. The EPIC model was used to simulate and calculate the crop growth indicators, and the results were fed back to the hydrologic model in HYDRUS to further simulate and analyze the soil water and salt transport. The model was encoded by FORTRAN 90 based on Microsoft Windows system with the actual evaporation and transpiration rates as upper boundary condition and subsurface drainage as the bottom boundary condition. The Penman–Monteith equation was used for the estimate of $ET_p$ and crop parameters, where the crop parameters adopted in this study was 1.45 [42,43].

### 2.2.1. Root Water Uptake

The root water absorption was calculated according to Feddes model as follows [44–48]:

$$S(h, h_\phi, z, t) = \alpha(h, h_\phi, z, t) \times b(z) \times T_p \tag{2}$$

where $S(h, h_\phi, z, t)$ is the root water absorption, $\alpha(h, h_\phi, z, t)$ is the stress response function, $b(z)$ is the distribution of water uptake, $mm^{-1}$, and $T_p$ is the potential crop transpiration, $mm\ d^{-1}$.

The actual transpiration rate $T_a$ was calculated by the following equation:

$$T_a = T_p \int_0^L \alpha(h, h_\phi) b(z) dz \tag{3}$$

where L is the root depth, cm, which was calculated as:

$$L_i = 2.5 L_{max}(HUI_i) \quad L_i \leq L_{max} \tag{4}$$

$$L_i = L_{max} \quad L_i > L_{max} \tag{5}$$

where $L_{max}$ is the maximum root depth, which was 80 cm in this study, and HUI is the heat unit coefficient of crop [42]. The root water uptake reduction factors are list in Table 3.

**Table 3.** Initial and calibrated values of water and salt stress function.

| Parameters (mm) | Description | Initial Values | Calibrated Values |
|---|---|---|---|
| $h_1$ | h when root can absorb water from soil (mm) | −150 | −180 |
| $h_2$ | h when root water absorption is not affected (mm) | −300 | −320 |
| $h_{3h}$ | h when WUR starts at high demand (mm) | −3250 | −3500 |
| $h_{3l}$ | h when WUR starts at low demand (mm) | −6000 | −6400 |
| $h_4$ | h when WUE = 0 (mm) | −80,000 | −85,000 |
| $h_{50\%}$ | h when WUR = 50% caused by salt stress (mm) | −50,000 | −55,000 |
| $T_h$ | Higher threshold of atmospheric evaporation capacity (mm $d^{-1}$) | 5 | 5 |
| $T_l$ | Lower threshold of atmospheric evaporation capacity (mm $d^{-1}$) | 1 | 1 |

Note: In this table, WUR refers to water uptake reduction.

### 2.2.2. Soil Solute Transport Module

The HYDRUS model uses convection dispersion equation to express soil solute transport process as follows:

$$\frac{\partial(\theta c + \rho s)}{\partial t} = \frac{\partial}{\partial z}\left(\theta D \frac{\partial c}{\partial z}\right) - \frac{\partial q c}{\partial z} \tag{6}$$

where c is the concentration of solute in soil solution, dS m$^{-1}$, $\rho$ is the dry bulk density of soil, g cm$^{-3}$, s is the concentration of solute adsorbed on soil particles, q is the Darcy velocity in direction, mm d$^{-1}$, and D is the hydrodynamic dispersion coefficient of saturated unsaturated soil, mm$^2$ d$^{-1}$.

### 2.2.3. Crop Growth Module

Generally, the harvest index value of most crops is relatively stable under stable environment and water and salt conditions [42]. Therefore, the model migrates the harvest index to calculate crop yield:

$$YLD = HIB_a \tag{7}$$

where YLD represents the crop yield, kg ha$^{-1}$, $B_a$ represents the aboveground biomass of the crop, kg ha$^{-1}$, and HI represents the harvest index of the crop, which was 0.45 in our study.

Aboveground biomass is affected by solar radiation, while solar radiation is closely related to the leaf area. Beer's law offers a calculation method as follows [45]:

$$\Delta B = (BE) \times 0.5(RA)[1 - \exp(-0.65LAI)] \tag{8}$$

where $\Delta$B is the potential increase in biomass, kg ha$^{-1}$, BE is the ability of crop convert solar energy into biomass, which was 25 (kg ha$^{-1}$) in the present study, RA is solar radiation, MJ m$^{-2}$, and LAI is the leaf area index [46,47].

### 2.2.4. Model Calibration and Validation

The measured and predicted soil water-salt content and crop grain yield of 2016 and 2017 were used to calibrate and validate the coupled model, respectively. The soil hydraulic and parameters and root water uptake and crop growth parameters were the main parameters calibrated and validated in this study. In our research, $\theta_s$ and $K_s$ were based on laboratory measurement, and $\theta_r$, n, a and root water uptake were the main calibration parameters since they were much more sensitive than $\theta_s$ and $K_s$ in the model operation process [31]. The initial input soil hydraulic parameters were predicted by van-Genuchten's model [41], and then calibrated and validated by the measured and predicted soil water content in 2016 and 2017, respectively. The root water uptake calculated according to Feddes model was calibrated and validated by the measured and predicted soil water-salt content in 2016 and 2017, respectively [44]. The initial crop growth parameters were estimated from the default values of the EPIC model [42], and then calibrated and validated by the measured and predicted grain yield data in 2016 and 2017, respectively.

### 2.2.5. Scenario Analysis

After calibration and validation, the coupled model was used to predict summer maize yield and water use efficiency (WUE) under different irrigation water salinity and drainage conditions. A total 49 simulation scenarios consisting of seven levels of irrigation water salinity (0, 2, 4, 6, 8, 10, and 12 dS m$^{-1}$) and seven levels of subsurface drain depth (no subsurface drain, 60, 80, 100, 120, 140, and 160 cm) were run with the validated coupled model. The input parameters such as soil physical properties, soil hydraulic parameters, and water uptake parameters were the same as the field experiment, and the daily meteorological data of 2016 were selected for the scenario simulation. To determine the effects of irrigation water salinity on relative grain yield and WUE under different drainage conditions, the polynomial regression method was used to fit the simulation results. Finally, the proper irrigation

water salinity was optimized based on the established relationships between grain relative yield and IWS under different drainage conditions.

## *2.3. Statistical Analysis*

The field observed soil water contents, soil salinities, ET, and grain yield obtained in 2016 and 2017 were used for calibrating and validating the coupled HYDRUS-EPIC model, respectively. The parameters root mean square error (RMSE), determination coefficient ($R^2$), and Nash–Sutcliffe efficiency (NSE) were adopted for evaluating model accuracy:

$$RMSE = \sqrt{\frac{1}{N} \sum_{i=1}^{N} (P_i - O_i)^2} \tag{9}$$

$$R^2 = \frac{\sum_{i=1}^{N} (O_i - O_a)(P_i - P_a)}{\sqrt{\sum_{i=1}^{N} (O_i - O_a)^2} \sqrt{\sum_{i=1}^{N} (P_i - P_a)^2}} \tag{10}$$

$$NSE = 1 - \frac{\sum_{i=1}^{N} (P_i - O_i)^2}{\sum_{i=1}^{N} (O_i - O_a)^2} \tag{11}$$

where RMSE is the root mean square error; $R^2$ is the determination coefficient; NSE is the Nash–Sutcliffe efficiency; $O_i$ is the observed value from the field experiment; $P_i$ is the predicted value from the model simulation; $O_a$ is the average of the observed value from the field experiment; $P_a$ is the average of the model simulation values. In general, the NSE ranges from $-\infty$ to 1. This value is close to 1, which is better model performance. $R^2$ ranges from 0 to 1, and RMSE values close to zero and $R^2$ close to 1 indicate good model performance.

## 3. Results

### *3.1. Model Calibration and Validation*

#### 3.1.1. Soil Water Contents

As shown in Figure 3a,b, field observed soil water contents within the soil depth 0~120 cm at the time of sowing (Figure 3a) and harvest (Figure 3b) in 2016 (calibration period) were compared with the simulated values. In our study, soil water content under high IWS was higher than that under low IWS in different drainage conditions. Moreover, the soil water contents were low at the drain depth since the subsurface drainage system provided an unobstructed drainage path. The differences between observed and predicted soil water contents at sowing time and harvest time were apparent during the calibration periods. The soil water content at harvest time was lower than that at sowing time. The main reason for this phenomenon was that the field was relatively dry at harvest and the average soil water content much lower. Therefore, the range of soil water content at harvest time was smaller than that at sowing time. However, the trend of soil water content at harvest time was similar to that at sowing time, i.e., it decreased with proximity to the drain depth. In D0, D1, and D2 at sowing time and harvest time, the experimental data and simulation data showed good correlation after calibration as shown in Figure 3. The model accuracy evaluating parameters shown in Table 4 indicated that, after calibration RMSE ranged from 0.02 to 0.05 $cm^3$ $cm^{-3}$, NSE ranged from 0.61 to 0.75, and $R^2$ ranged from 0.68 to 0.85. As shown in Figure 4, during validation (2017), the experimental data and simulation data also show good correlation relationship, since both were distributed near the 1:1 line. As shown in Table 4, RMSE during validation ranged from 0.01 to 0.03 $cm^3$ $cm^{-3}$, NSE ranged from 0.63 to 0.75, and $R^2$ ranged from 0.71 to 0.88, and the floating range of these parameters were similar compared with other studies [31,49]. By comparing the observed and predicted soil water content data shown in Figures 3 and 4 and evaluating the model accuracy parameters listed in Table 4, it was generally believed that the soil hydraulic parameters in Table 2 were acceptable for the coupled

model. Thus, we confirmed that the simulation effect was satisfactory in soil water content simulation under HYDRUS-EPIC model.

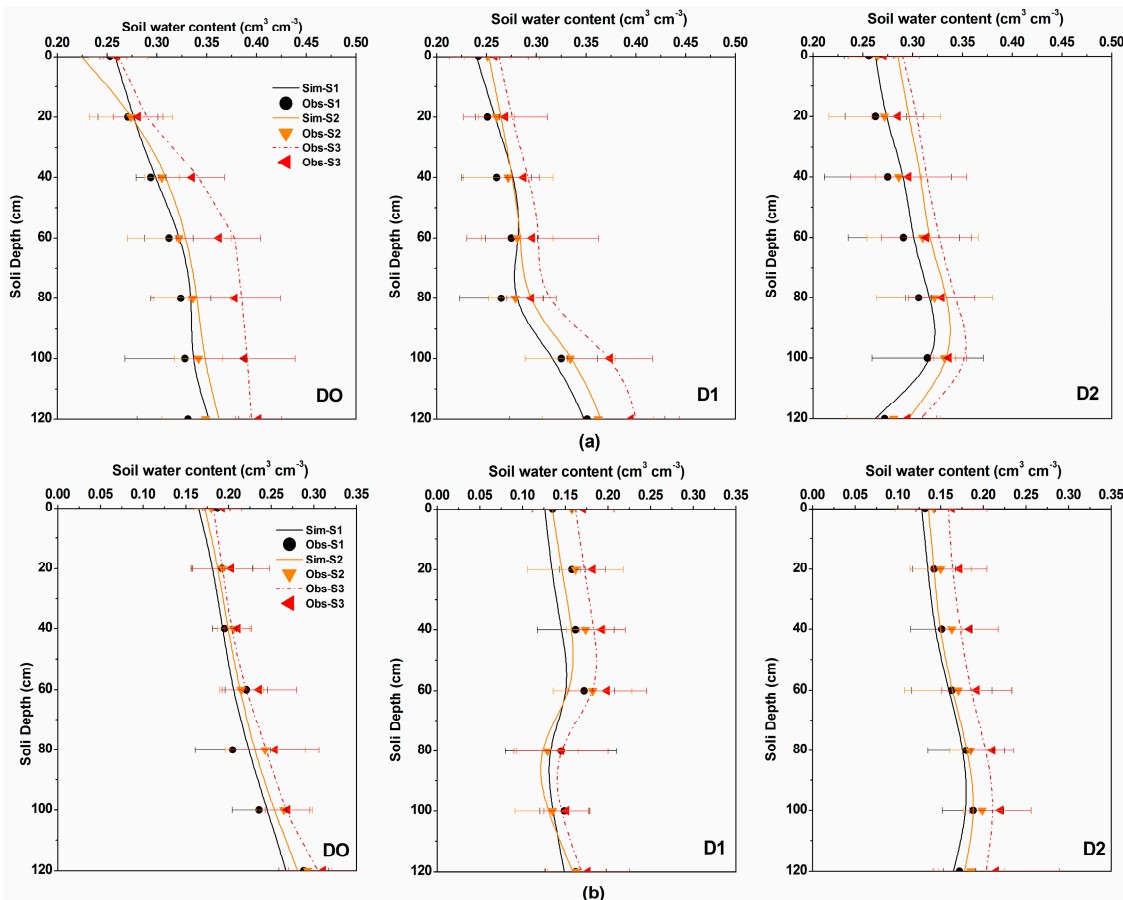

**Figure 3.** The observed and predicted soil water contents at sowing time (**a**) and harvest time (**b**) during calibration (2016).

**Table 4.** Model accuracy evaluation for soil water-salt dynamics in 2016 and 2017.

| Year | Drain Depth (cm) | Soil Water Content (cm³cm⁻³) | | | EC_e (dS m⁻¹) | | |
|---|---|---|---|---|---|---|---|
| | | RMSE | NSE | $R^2$ | RMSE | NSE | $R^2$ |
| 2016 (Calibration) | D0 (no drain) | 0.03 | 0.61 | 0.75 | 0.28 | 0.68 | 0.83 |
| | D1(80) | 0.02 | 0.63 | 0.79 | 0.39 | 0.51 | 0.71 |
| | D2 (120) | 0.02 | 0.71 | 0.85 | 0.29 | 0.58 | 0.75 |
| 2017 (Validation) | D0 (no drain) | 0.03 | 0.63 | 0.71 | 0.31 | 0.66 | 0.85 |
| | D1 (80) | 0.03 | 0.68 | 0.80 | 0.48 | 0.51 | 0.65 |
| | D2 (120) | 0.02 | 0.75 | 0.88 | 0.35 | 0.54 | 0.75 |

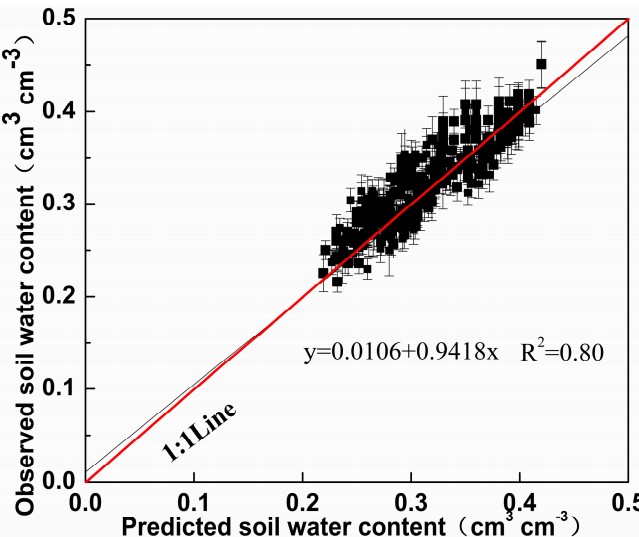

**Figure 4.** The observed and predicted soil water contents during validation (2017).

### 3.1.2. Soil Salinity

Figure 5 shows a comparison between the field observed soil salinity data and the simulated data within the root zone depth in 2016 (calibration period). Compared to D0 treatment, the increase of $EC_e$ was slight for S1 (1.95~2.55 dS m$^{-1}$), and more striking for S2 (2.48~3.45 dS m$^{-1}$) and S3 (2.58~3.95 dS m$^{-1}$). The reason for this phenomenon is that there was no drainage under this treatment, therefore the soil salinity in S2 and S3 fluctuated with irrigation according to the irrigation water salinity. At the drain depths of 80 cm (D1) and 120cm (D2), similar $EC_e$ dynamics are observed shared with the three levels of IWS (S1, S2, and S3), where $EC_e$ increased with increasing IWS. Moreover, $EC_e$ was lower when SDD was 80 cm (D1) than when it was 120 cm (D2).

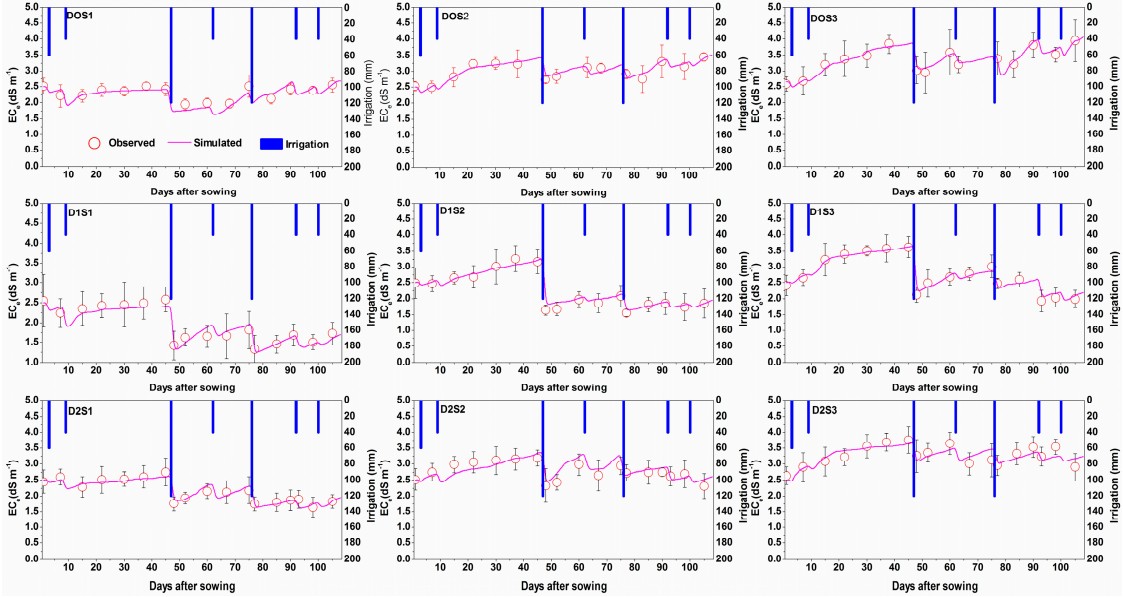

**Figure 5.** The observed and predicted $EC_e$ within the root zone depth during calibration (2016).

In D0, D1, and D2 treatments, the experimental data and simulation data showed good correlation during calibration, as shown in Figure 5. The model evaluating parameters shown in Table 4 also indicated that RMSE during calibration (2016) ranged from 0.28 to 0.39 dS m$^{-1}$, NSE ranged from 0.51to 0.68, and $R^2$ ranged from 0.71 to 0.83. As shown in Figure 6, during validation (2017), the experimental data and simulation data also incarnated great correlation relationship, since both forms of data

were distributed near the 1:1 line. As shown in Table 4, RMSE in the validation period ranged from 0.31 to 0.48 dS m$^{-1}$, NSE ranged from 0.51 to 0.66, and R$^2$ ranged from 0.65 to 0.85. In this research, the NSE and R$^2$ values for D1 and D2 were lower than that for D0 in the simulation soil salinity. Similar simulation results were obtained by Ren et al. who obtained R$^2$ values in the range of 0.65–0.91 [50]. By comparing the observed and predicted soil salinity data shown in Figures 5 and 6 and evaluating the model accuracy parameters listed in Table 4, it was generally believed that the soil hydraulic parameters in Table 2 and water-salt stress function in Table 3 were acceptable for the coupled model. Thus, we confirmed that the simulation effect was satisfactory in soil salinity simulation under HYDRUS-EPIC model.

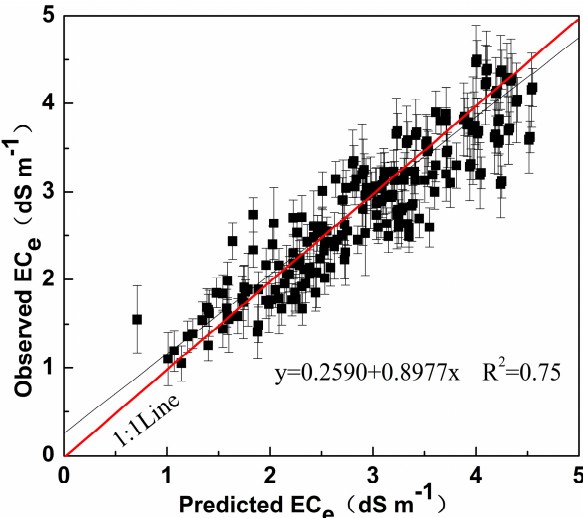

**Figure 6.** The observed and predicted EC$_e$ within the root zone depth during validation (2017).

### 3.1.3. ET and Crop Yield

As shown in Table 5, the measured ET values were generally lower than the predicted ET values, but the experimental data and simulation data showed good correlation during calibration (2016). During calibration, the RMSE, NSE, and R$^2$ were 7.65 mm, 0.95 and 0.98, respectively (Table 5), while during validation (2017), the RMSE, NSE, and R$^2$ were 11.30 mm, 0.92 and 0.97, respectively (Table 5). As shown in Table 5, the measured grain yield values were generally lower than the predicted grain yield values in both 2016 and 2017 growing seasons. The grain yield decreased with increasing irrigation water salinity under the same drainage conditions. Moreover, the mean grain yield for D1 (drain depth: 80 cm) was obviously higher than that for D0 (no subsurface drainage) and D2 (drain depth: 120 cm) under the same irrigation water salinity. The highest ET values were obtained in D0S1 in both by observed and by simulation in 2016 and 2017 growing seasons. The ET values were obtained by calculating the water balance process in both the field experiments and the HYDRUS model. For treatments D1 and D2, discharge in the water balance was excluded by subsurface drainage effects, and thus the ET values for D0 were higher than for D1 and D2. Furthermore, the highest grain yield values were obtained in D1S1 by both observed and by simulation in 2016 and 2017 growing seasons. As was discussed above, the IWS of 0.25 dS m$^{-1}$ (S1) introduced the least amount of salt in the soil profile which minimized salt stress on the crop compared with IWS of 3.75 (S2) and 6.25 dS m$^{-1}$ (S3). Moreover, the drainage depth D1 better reduced the salt content within the crop root zone compared to drainage depth of D0 and D2. During calibration (2016), the RMSE, NSE, and R$^2$ for grain yield simulation were 255.71 kg ha$^{-1}$, 0.85, and 0.98, respectively (Table 5), while during validation (2017), the RMSE, NSE, and R$^2$ were 212.21 kg ha$^{-1}$, 0.87, and 0.94, respectively (Table 5). Similar research have evaluated winter wheat growth and yield in the Loess Plateau of China using the EPIC model [51,52]. Based on Table 5, ET and grain yield simulation by the HYDRUS-EPIC model was deemed satisfactory.

**Table 5.** Simulated and observed summer maize evapotranspiration, yields.

| Yea· | Treatments | Evapotranspiration (mm) | | | | | Grain Yields (kg ha$^{-1}$) | | | | |
|---|---|---|---|---|---|---|---|---|---|---|---|
| | | Observed | Simulated | RMSE | NSE | $R^2$ | Observed | Simulated | RMSE | NSE | $R^2$ |
| 2016 (Calibration) | D0S1 | 513 ± 12.8 | 502.5 | | | | 7123 ± 205 | 7342.3 | | | |
| | D0S2 | 488 ± 15.2 | 479.8 | | | | 6223 ± 156 | 5823.5 | | | |
| | D0S3 | 463 ± 11.0 | 462.2 | | | | 5745 ± 134 | 5615.6 | | | |
| | D1S1 | 425 ± 13.2 | 435.6 | | | | 7911 ± 336 | 7854.8 | | | |
| | D1S2 | 405 ± 5.3 | 400.2 | 7.65 | 0.95 | 0.98 | 7358 ± 105 | 7156.5 | 255.71 | 0.85 | 0.97 |
| | D1S3 | 399 ± 10.1 | 398.4 | | | | 6899 ± 126 | 6715.8 | | | |
| | D2S1 | 458 ± 13.6 | 466.6 | | | | 7563 ± 121 | 7236.4 | | | |
| | D2S2 | 447 ± 11.5 | 455.5 | | | | 6626 ± 213 | 6328.3 | | | |
| | D2S3 | 440 ± 11.2 | 448.2 | | | | 6218 ± 212 | 5915.6 | | | |
| 2017 (Validation) | D0S1 | 493 ± 12.8 | 495.6 | | | | 7224 ± 115 | 7412.9 | | | |
| | D0S2 | 471 ± 12.1 | 482.3 | | | | 6188 ± 186 | 6434.6 | | | |
| | D0S3 | 460 ± 13.0 | 472.6 | | | | 5442 ± 120 | 5052.5 | | | |
| | D1S1 | 400 ± 10.3 | 409.3 | | | | 8325 ± 163 | 8115.3 | | | |
| | D1S2 | 392 ± 13.4 | 395.6 | 11.30 | 0.92 | 0.97 | 7702 ± 158 | 7552.5 | 212.21 | 0.87 | 0.94 |
| | D1S3 | 385 ± 12.8 | 392.2 | | | | 7236 ± 136 | 7455.8 | | | |
| | D2S1 | 483 ± 19.9 | 492.5 | | | | 7725 ± 110 | 7913.5 | | | |
| | D2S2 | 472 ± 20.6 | 480.6 | | | | 6842 ± 150 | 6782.1 | | | |
| | D2S3 | 447 ± 16.2 | 449.9 | | | | 6326 ± 158 | 6383.8 | | | |

### 3.2. Effects on Relative Grain Yield and WUE

A contour map of the relative grain yield in each simulation scenario was generated as shown in Figure 7. When there is no subsurface drainage system, the relative yield of summer maize is about 0.5 at IWS of 8 dS m$^{-1}$. Additionally, when IWS was 12 dS m$^{-1}$, the relative yield of summer maize is 0, which means that summer maize cannot survive. Under drain depth of 100 cm, the relative yield of summer maize is about 0.82 at IWS of 8 dS m$^{-1}$. While at IWS of 12 dS m$^{-1}$, the relative yield of summer maize is 0.53, which was much higher than that under no subsurface drainage system. Under drain depth of 160 cm, the relative yield of summer maize is about 0.53 at IWS of 8 dS m$^{-1}$. While at IWS of 12 dS m$^{-1}$, the relative yield of summer maize is just 0.23, which is much lower than that under SDD of 100 cm. The results above indicated that grain yield reduction could be effectively restrained by proper arrangement of subsurface drainage system even under saline water irrigation of high concentration. Figure 7 also reflects that the relative grain yield increases initially and then decreases with the increase of SDD under all levels of IWS. In this study, the relative grain yields obtained peak values when drain depth was around 90~100 cm.

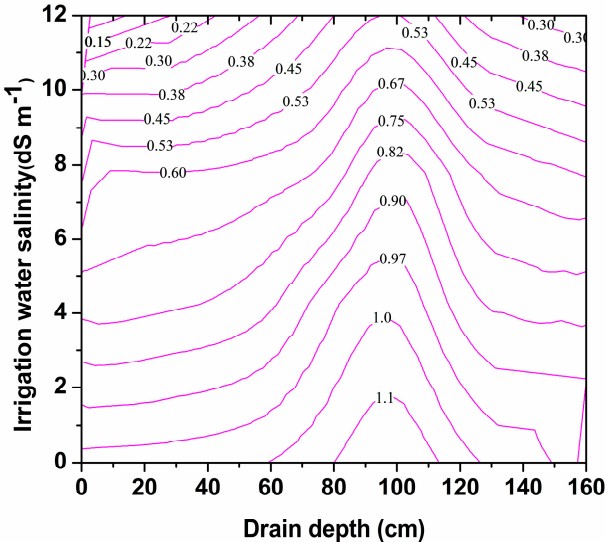

**Figure 7.** Contour map of the relative grain yield in each simulation scenario.

The ET in each scenario was also simulated with the coupled HYDRUS-EPIC model, and the polynomial regression method was used to fit the calculated WUE, and then the contour map of the WUE in each simulation scenario was generated as shown in Figure 8. As shown in Figure 8, the WUE decreased with increasing IWS. The highest WUE was obtained in the scenario with SDD of 100 cm, and IWS of 0 dS m$^{-1}$, where the WUE reached 2.08 kg m$^{-3}$. The lowest WUE was obtained in scenario with no subsurface drainage, and the IWS of 12 dS m$^{-1}$, where the WUE was 0 kg m$^{-3}$, since the relative grain yield was 0 in this scenario. From the overall perspective of Figure 8, with a constant subsurface drainage condition, the highest mean WUE was realized in scenarios where IWS was 0 dS m$^{-1}$, and the mean WUE was 1.64 kg m$^{-3}$. Additionally, the lowest mean WUE was realized in scenarios where IWS was 12 dS m$^{-1}$, and the mean WUE was 0.55 kg m$^{-3}$. Figure 8 also indicated that, with the increase of SDD, the WUE increased first and then decreased at the SDD of 100 cm. The mean WUE for SDD of 100 cm was 1.66 kg m$^{-3}$. The lowest mean WUE was in the scenarios where there was no subsurface drainage condition, and the value was just 0.88 kg m$^{-3}$. Consequently, both the IWS and SDD were the main factors that affected the crop yield and water use efficiency of summer maize.

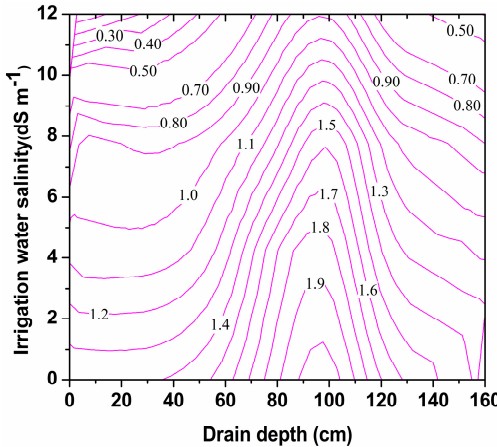

**Figure 8.** Contour map of the water use efficiency (WUE) in each simulation scenario.

### 3.3. Optimizing IWS under Subsurface Drainage Condition

As shown in Table 6, there were significant linear relationships between relative grain yield and IWS under different drainage conditions, where the determination coefficients $R^2$ were greater than 0.95. Under drain depth of 100 and 120 cm, the relative grain yield decreased the least (5.25% and 5.93%, respectively) with increasing IWS. While under drain depth of 160 cm and no subsurface drainage treatments, the relative grain yield decreased the most (6.84% and 6.91%, respectively) with increasing IWS. It should be noted that the threshold values of IWS when summer maize start reducing grain yield production ($Y_r$ = 100%) were different under each drain depth. The highest threshold value of IWS under SDD of 80 and 100 cm were 2.89 and 4.81 dS m$^{-1}$, respectively. Again, this result confirmed that the effect of salt stress on crop growth was weakened under SDD of 80 and 100 cm. Thus, even under relative higher irrigation water salinity, the growth of corn was not affected obviously. While under SDD of 140, 160, cm and no subsurface drainage condition, the threshold values of IWS were just 0.70, 0.32, and 0.42 dS m$^{-1}$. The threshold values of IWS when summer maize yield reduction reaches 20% ($Y_r$ = 80%) were also different in each simulation scenario. Under drain depth of 80 and 100 cm, the threshold values of IWS were relatively high, which were 6.19 and 8.61 dS m$^{-1}$, respectively, while under drain depth of 140, 160 cm and no subsurface drainage condition, the threshold values were much lower than in the other treatments, and they were 3.98, 3.70, and 3.35 dS m$^{-1}$, respectively. The simulation results above indicated that under drain depth of about 100 cm, the sensitivity of relative grain yield to salt concentration of irrigation water was relatively low. Therefore, based on the results of the scenario simulations and data analysis, it is considered that subsurface drainage condition can provide considerable potential for the sustainable utilization of saline water irrigation.

**Table 6.** Relationship between relative grain yield and irrigation water salinity (IWS) under different drainage conditions.

| Drain Depth (cm) | The Decline Rate [1] (%) | $EC_{iw}$ [2] (dS m$^{-1}$) | | | | $R^2$ |
|---|---|---|---|---|---|---|
| | | $Y_r$ = 100% | $Y_r$ = 80% | $Y_r$ = 50% | $Y_r$ = 0% | |
| No subsurface drainage | 6.84 | 0.42 | 3.35 | 7.73 | 15.04 | 0.96 |
| 60 | 6.04 | 1.31 | 4.62 | 9.59 | 17.87 | 0.99 |
| 80 | 6.05 | 2.89 | 6.19 | 11.15 | 19.41 | 0.97 |
| 100 | 5.25 | 4.81 | 8.61 | 14.33 | 22.98 | 0.95 |
| 120 | 5.93 | 1.54 | 4.91 | 9.97 | 18.40 | 0.96 |
| 140 | 6.09 | 0.70 | 3.98 | 8.91 | 17.12 | 0.97 |
| 160 | 6.91 | 0.32 | 3.70 | 8.78 | 17.24 | 0.96 |

Note: 1. The decline rate refers to the decline rate of relative yield when irrigation water salinity increased by 1dS m$^{-1}$; 2. The $EC_{iw}$ refers to the salinity of irrigation water.

## 4. Discussion

### 4.1. Effects on Soil Salinity

In the two-year field experiment, the soil $EC_e$ within the root zone was significantly reduced during the leaching period: Forty eight days after sowing and 78 days after sowing as shown in Figure 5. The decline range under D0 was much lower than that under other drainage conditions. This phenomenon was caused by the subsurface drainage systems which were installed at different soil depths. Similar studies by Bahceci et al. and Lu et al. suggested that the $EC_e$ within the root zone would reach a relative high level with the increase of IWS [1,53]. The proper depth of subsurface drainage changes with soil properties, soil salinity, and irrigation water salinity [54,55]. Using SALTMOD, Srinivasulu et al. and Ghumman reported that it is not necessary to bury the subsurface pipe to a great depth, when the SDD > 1.4 m, the control of soil water-salt by subsurface drainage would not be meaningful [17]. In our study, simulation results indicated that with SDD around 80~100 cm, $EC_e$ within the root zone could be controlled to a proper level even under relatively high IWS. Therefore, both field experiment and simulation results obtained in this study proved that subsurface drainage condition could control soil water-salt within the root zone with a proper SDD. Moreover, our study confirmed that when subsurface drainage condition is adopted in farmland soil water-salt control, both field experiments and numerical model are essential for evaluating the sustainable utilization of saline water.

### 4.2. Improved Grain Yield

The grain yield reduction for summer maize was influenced by the root water uptake process, which was affected by water-salt stress within the root zone depth [56]. In the present study, relative grain yield was about 0.80 when IWS was 8.61 dS m$^{-1}$ and SDD was 100 cm. This result suggested that during the critical growth periods of summer maize, the necessary salt leaching process of the root zone was beneficial for the crops to avoid salt stress. As was reported in previous studies, maize as a moderately salt sensitive crop, when the soil salinity exceeds about 2 dS m$^{-1}$, the grain yield reduction occurred [57]. Other previous studies also indicated that the reduction rate of grain yield could be retarded with proper irrigation and drainage systems [45,49]. Similar research by Russo indicated that the grain yield reductions were about 8 %and 27%, respectively, at the IWS of 3.6 and 6.7 dS m$^{-1}$, respectively [58]. However, in their experiment, there was no condition for the salt leaching process during the crop growing seasons. In our study, the reduction rates were much lower, which was about 5.25%–6.91% for every 1 dS m$^{-1}$ increase in IWS under subsurface drainage condition. Our research confirmed that under proper SDD, the utilization potentiality of saline water irrigation would be remarkably enhanced.

### 4.3. Ability of the Coupled Model

In our research, the coupled model HYDRUS-EPIC model was used to evaluate the sustainable use of saline water irrigation under subsurface drainage systems. In the field experiment design, the saline water irrigation and subsurface drainage measure were main boundary conditions that affected soil water-salt movement process during the simulation periods. As was reported by previous study, with multiple boundary conditions provided by HYDRUS model, the application ability of HYDRUS model was strengthened [20,21]. With the same model, Tao et al. evaluated the performance of the improved subsurface drainage in Huaibei plain by field experiment and numerical simulation [39]. Ebrahimian and Noory applied HYDRUS-2D model to simulate water flow under subsurface drainage in a paddy field for various drain depths and spacing, surface soil textures, and crack conditions [27]. In crop growth simulation, the EPIC crop growth module used a unified approach to simulate more than 100 types of crops with less input data [43]. When coupled with HYDRUS model, the coupled model had strong advantages in simulating soil hydrological process and crop growth under various condition and crop types [51,52]. In our study, the outputs of current study could be adopted in the

design of irrigation schedule under different hydrological year and soil water-salt environment in the future.

## 5. Conclusions

In the present study, a coupled hydrologic process and crop growth model was evaluated using field experiment data during 2016 and 2017. The coupled model was run to simulate the effects of different IWS and SDD on soil water-salt dynamics, ET, and grain yield of summer maize. The coupled HYDRUS-EPIC model was calibrated and validated adequately by comparing the measured and predicted values from the two growing seasons. A seven × seven scenario simulation was conducted to evaluate the effects of IWS and SDD on relative grain yield and WUE. The results indicated that grain yield reduction could be effectively restrained by the proper arrangement of subsurface drainage system even under high concentration of IWS. Under SDD of 100 cm, the relative yield of summer maize was about 0.82 at IWS of 8 dS $m^{-1}$ and about 0.53 at IWS of 12 dS $m^{-1}$. The highest WUE was obtained in the scenario with SDD of 100 cm, and IWS of 0 dS $m^{-1}$, where WUE reached 2.08 kg $m^{-3}$. A significant linear correlation was found between the relative grain yield of summer maize and the IWS under each drainage condition ($R^2 > 0.95$). The optimization results indicated that under drain depth of about 100 cm, the sensitivity of relative grain yield to salt concentration of irrigation water was relatively low in this study. Even when the IWS was as high as 8.61 dS $m^{-1}$, the yield reduction of summer maize was still less than 20%. In summary, the research indicated that subsurface drainage measures may provide important support for the sustainable utilization of saline water. Moreover, the coupled HYDRUS-EPIC model should be a beneficial tool to evaluate the future sustainability of the saline water irrigation systems.

**Author Contributions:** G.F. and Z.Z. (Zhanyu Zhang) designed this study. G.F. and Z.Z. (Zemin Zhang) contributed to field experiment and data processing. G.F. developed and operated the HYDRUS-EPIC model. G.F. prepared the original Draft. Z.Z. (Zhanyu Zhang) reviewed and edited the manuscript.

**Funding:** This research was funded by the China Postdoctoral Science Foundation (2017M621619); the Fundamental Research Funds for the Central Universities (2019B117114, 2018B34014), and the Natural Science Foundation of China (Granted No. 51879071).

**Acknowledgments:** We are grateful to the anonymous reviewers and editor for their insightful comments and suggestions on this paper. We sincerely thank Ahmad Bakour and Richwell Mubita Mwiya for their language assistance in this paper.

**Conflicts of Interest:** The authors declare no conflict of interest.

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
