# Peer review of "Evaluating the Sustainable Use of Saline Water Irrigation on Soil Water-Salt Content and Grain Yield under Subsurface Drainage Condition"

_sustainability, doi:10.3390/su11226431_

Round 1
Reviewer 1 Report
I appreciate the authors to address the comments.
Reviewer 2 Report
I am satisfied about the work carried out by the authors in revising the paper.
This manuscript is a resubmission of an earlier submission. The following is a list of the peer review reports and author responses from that submission.
Round 1
Reviewer 1 Report
I appreciate the Editor to give me a chance to review an interesting and valuable paper. I found some merits in the both methodology and results. In my opinion, this paper has a good potential to be published in the journal. However, I have also some concerns on the different parts of the manuscript. If the author(s) address carefully to the comments, I’ll recommend publication of the manuscript in the journal:
Discuss more the main reasons for the differences between observed and predicted soil water contents at sowing time and harvest time during the calibration periods.
Add regression equation and R-squared for observed and predicted soil water contents during the validation periods and observed and predicted ECe within the root zone depth during the validation periods.
In Tables 5 and 6, highlight values that are more important and discuss them for better understanding readers.
Focus on the ability of HYDRUS in peak and low point events during the experiment.
How can extend the results in other regions with similar/different soil condition?
At the end of the manuscript, explain the implications and future works considering the outputs of current study.
The quality of the language needs to improve by a native English speaker for grammatically style and word use.
Reviewer 2 Report
The authors propose a study where they use two coupled models to evaluate the effects of irrigation with saline water of maize. The topic is interesting, but in my honest opinion the paper has many flaws, that hampers its publicatin in the present form. Below my major comments:
in the Introduction the authors dis not explain (or did not support with literatire evidences) the novelty of their study
the Methods have a unusual structure and lack many information (mainly about the model implemetation)
the Results did not show separately model calibration and validation outcomes
many methodological information is reported in Results
the MS is plenty of typos and language errors
English deserves a hard revision.
Other comments are reported in the annotated MS.
